# Cyclodextrin-Based Nanoparticles for Delivery of Antisense Oligonucleotides Targeting Huntingtin

**DOI:** 10.3390/pharmaceutics15020520

**Published:** 2023-02-03

**Authors:** Monique C. P. Mendonça, Yao Sun, Michael F. Cronin, Andrew J. Lindsay, John F. Cryan, Caitriona M. O’Driscoll

**Affiliations:** 1Phamacodelivery Group, School of Pharmacy, University College Cork, T12 YT20 Cork, Ireland; 2Membrane Trafficking & Disease Laboratory, School of Biochemistry & Cell Biology, Biosciences Institute, University College Cork, T12 YT20 Cork, Ireland; 3APC Microbiome Ireland, University College Cork, T12 YT20 Cork, Ireland; 4Department of Anatomy and Neuroscience, University College Cork, T12 XF62 Cork, Ireland

**Keywords:** Huntington’s disease, nanomaterials, ASO, brain delivery

## Abstract

Huntington’s disease (HD) is a progressive inherited neurodegenerative disease caused by a CAG repeat expansion in the huntingtin gene, which is translated into the pathologic mutant huntingtin (mHTT) protein. Despite the great potential of HTT lowering strategies and the numerous antisense oligonucleotides (ASOs) in pre- and clinical trials, sustained silencing of mHTT has not been achieved. As a strategy to improve ASO delivery, cyclodextrin-based nanoparticles (CDs) offer a promising approach. Here, three CDs with distinct chemical structures were designed and their efficacies were compared as potential platforms for the delivery of ASO targeting HTT. Results using striatal neurons and HD patient-derived fibroblasts indicate that modified γ-CDs exhibited the best uptake efficiency and successfully downregulated mHTT at protein and allele levels. The incorporation of the brain-targeting peptide RVG into the modified γ-CDs showed greater downregulation of mHTT protein and HD-causing allele SNP1 than untargeted ones in an in vitro blood–brain barrier model. Although the ASO sequence was designed as a nonallele-specific therapeutic approach, our strategy gives an additional benefit of some mHTT selectivity. Overall, this study demonstrated the CD platform’s feasibility for delivering ASO-based therapeutics for HD treatment.

## 1. Introduction

Huntington’s disease (HD) is a progressive, inherited neurodegenerative disorder caused by a CAG repeat expansion in the huntingtin gene, which results in the production of mutant huntingtin (mHTT) protein with an abnormally long polyglutamine (polyQ) expansion (≥36 CAG repeats). This mutant form of the HTT protein acquires a toxic gain-of-function by forming self-aggregates, leading to neuronal dysfunction and, ultimately, cell death [1]. Given that the pathology is driven by the toxic mHTT protein, therapeutic strategies focused on lowering mHTT levels have been developed through the years. These strategies mainly include antisense oligonucleotides (ASO) and RNA interference (RNAi) based technologies, which target the RNA product in an allele-selective or non-selective way [2,3].

Ubiquitous at the tissue and subcellular levels, wild-type HTT (wtHTT) plays an essential role in embryonic neurodevelopment and has numerous functions in the adult brain, including transcriptional regulation, vesicular transport, regulation of endocytosis and vesicular trafficking, cellular stress responses (survival and cell death), and DNA repair [4]. The long-term consequences of wtHTT reduction in the adult brain remain unclear. For this reason, selective reduction of mHTT while maintaining wtHTT levels may be a safer approach in the longer term than the unselective knockdown of both HTT alleles [5,6]. Although not covering all HD patients, approximately two-thirds of HD patients with European ancestry are expected to possess single-nucleotide polymorphism (SNP) rs362307 (SNP1), rs362331 (SNP2), or both SNPs in association with the CAG repeat expansion, making these SNP variants promising therapeutic targets for the majority of HD patients [7].

Despite initial promising outcomes, three clinical trials using HTT-lowering ASOs were terminated in 2021: the non-allele selective GENERATION HD1 (NCT03761849) trial, and the two distinct allele-selective PRECISION-HD1 (NCT03225833) and PRECISION-HD2 (NCT03225846) trials. The GENERATION HD1 trial aimed to evaluate the efficacy and safety of intrathecal tominersen in adults. Although there were significant decreases in cerebrospinal fluid (CSF) mHTT, patients treated with tominersen fared worse than those in the placebo group in Phase III. Wave Life Sciences’ PRECISION-HD trials aimed to evaluate the efficacy and safety of intrathecal WVE-120201 and WVE-120202, which carried SNPs rs362307 and rs362331, respectively. Preliminary results of both trials did not show significant decreases in mHTT or total HTT at the doses tested [8]. Given these failures, the exploration of alternative ways to optimize ASO drug delivery has become a research hotspot.

Different strategies have been developed to improve ASO stability, targeting, and delivery, including chemical modifications, bioconjugation to different moieties, and the use of delivery vehicles [9]. Among the delivery vehicles, cyclodextrins (CDs) are known as favorable natural nanocarriers for oligonucleotide therapy [10,11,12,13]. Cyclodextrins are non-toxic cyclic oligosaccharides constituted by six (α-CD), seven (β-CD), and eight (γ-CD) d-glucopyranosyl units. They can be synthesized in relatively high quantities and exhibit a remarkable versatility for chemical modification, allowing for the addition of unique functional groups for cell and tissue-specific targeting [14]. The conjugation of small peptides is especially interesting in this respect.

In previous studies, we developed new CD nanoplatforms based on PEGylated amphiphilic CDs conjugated with the brain-targeting peptide rabies virus glycoprotein (RVG) covalently via thiol–maleimide click chemistry [15] and by CD–adamantyl inclusion complex formation [16]. Regardless of the chemistry approach, both CDs successfully increased the delivery of siRNA into neuronal cells and induced gene knockdown. RVG can specifically interact with neuronal nicotinic acetylcholine receptors (nAChR) or γ-aminobutyric acid (GABA) receptors on the surface of endothelial cells and transport a variety of nanocarriers into the central nervous system (CNS) in a non-invasive way [17].

In the present work, we designed and characterized three CDs with distinct chemical structures and properties as potential platforms for the delivery of ASO targeting total HTT. These CDs:ASO nanocomplexes were screened for cellular uptake and their potential to induce HTT protein knockdown. The lead formulation was then conjugated to RVG by the post-insertion method in order to improve brain uptake and silencing efficacy. The efficiency of the RVG-targeted CDs:ASO nanocomplexes was evaluated in an in vitro co-culture model of the blood–brain barrier (BBB), consisting of the human cerebral microvascular endothelial cell line (hCMEC/D3) and the rat striatal embryonic neuronal cell line, ST14A, which expresses human mHTT (residues 1-548) with 120 CAG repeats and has been established as a reliable model of HD [18,19]. Finally, RVG-targeted CDs:ASO nanocomplexes were validated in a patient-derived cellular model of HD containing clinically relevant CAG expansion-associated SNPs.

## 2. Materials and Methods

### 2.1. Oligonucleotide

The human HTT-targeting ASO was purchased from Eurogentec (Seraing, Belgium) and was 20 nucleotides in length with five 2′-O-methoxyethyl (2′-O-MOE) modified nucleosides at the 5′ and 3′ends (upper case letters) and ten unmodified nucleotides in the central part (lower case letters). The internucleoside bonds (marked with an asterisk) were phosphorothioated to improve nuclease resistance. ASO sequence was as follows: C*TCAG*t*a*a*c*a*t*t*g*a*c*ACCA*C*. This sequence was previously described by [20,21].

### 2.2. Cyclodextrins-Based NPs

#### 2.2.1. Cyclodextrins:ASO Nanocomplexes

Three different CDs were investigated (Figure 1): Cationic amphiphilic β-CDs synthesized as previously described [22] using the “click chemistry” strategy, herein referred to as CD1; a second cationic amphiphilic β-CDs, previously described [16] and designed with an amino group to decrease the pKa relative to CD1, herein referred to as CD2; and a γ-CD with a double-charged chain synthesized and kindly provided by Cyclolab, R&D Laboratory (Hungary), herein referred to as CD3. These structures were selected to compare the type of CDs-β-CDs (CD1 and CD2) vs. γ-CDs (CD3), the cationic nature of the headgroup—primary amines (CD1) vs. tertiary amines (CD2), and the presence of a second charged chain (CD3, primary amines). 

The CDs were reconstituted in RNAse-free water to obtain a 0.25 mg/mL (CD1 and CD2) or 1 mg/mL (CD3) solution. These solutions were mixed with ASO (500 nM) at different weight ratios (WR) of CDs:ASO (5:1; 7.5:1 and 10:1) and incubated at 65 °C and 450 rpm for 30 min in a thermomixer.

#### 2.2.2. Synthesis of DSPE-PEG-RVG

For surface modification of the CDs:ASO nanocomplexes, DSPE-PEG_2000_-Maleimide (DSPE-PEG-Mal) was conjugated to RVG by the Michael reaction. Firstly, the -SH group in RVG peptide (N’-CYTIWMPENPRPGTPCDIFTNSRG KRASNGGGG-C′, TAG Copenhagen, Denmark, Belgium) was activated with 2-Mercaptoethanol (2-ME, M3148, Sigma-Aldrich, Wicklow, Ireland) by mixing RVG (1 Equiv) and 2-ME (15 Equiv) in deionized water (DIW) for 4 h with continuous stirring at room temperature and under nitrogen. The final product was then freeze-dried. The activated RVG peptide (1 Equiv) was dissolved in a solvent mixture of chloroform, methanol, and DIW [CHCl3:MeOH:DIW = 65:35:8 (*v*/*v*/*v*)] and added to the DSPE-PEG-Mal (1.5 Equiv). Using triethylamine as a catalyst, the solution was incubated at 41 °C for 48 h with stirring, avoiding oxygen and light. Post incubation, the solution was dialyzed against 96% ethanol in a dialysis cassette (3.5k MWCO, 66,332, Thermo Fisher Scientific, Bremen, Germany) for 2 days, replacing 96% ethanol every 12 h to remove triethylamine and any unconjugated DSPE-PEG-Mal. The product, named DSPE-PEG-RVG, was freeze-dried and stored at −20 °C until used. The DSPE-PEG-Mal-RVG product was identified by ^1^H NMR (Appendix A).

To prepare non-targeted and RVG-targeted formulations, DSPE-PEG/DSPE-PEG-RVG was dissolved in water (1 mg/mL) and sonicated for 30 min in an ultrasonic bath at 60 °C. Subsequently, DSPE-PEG/DSPE-PEG-RVG was added to the CDs:ASO nanocomplexes (WR 10:1, 750 nM) in order to achieve CDs:ASO:DSPE-PEG (0.35 molar ratio of DSPE-PEG to CD) or CDs:ASO:DSPE-PEG-RVG (60 mol% to DSPE-PEG). The non-targeted and target formulations were incubated at 65 °C and 450 rpm for 30 min in a thermomixer.

### 2.3. Physicochemical Profiles

Hydrodynamic size, polydispersity index (PDI), and zeta potential were determined by dynamic light scattering using a Zetasizer NanoZS^®^ instrument (Malvern Instruments, Worcestershire, UK). The size and morphology were verified by transmission electron microscope (TEM). The samples were deposited to a 400-mesh carbon film copper grid, stained with 2% (*w*/*v*) phosphotungstic acid, and imaged at a JEOL operated at 200 kV. The binding of ASO to CDs was confirmed by agarose gel electrophoresis, as previously described [16].

Heparin displacement was conducted to study the release of ASO from CDs. Heparin (10 mg/mL) was added to the complexes (WR of 1:5) for 30 min, 6 h, 24 h, 48 h, and 72 h, and agarose gel electrophoresis was performed at 90 V for 60 min. Finally, the stability profile in a simulated biological environment was evaluated by incubating the CDs:ASO nanocomplexes with an equal volume of 25% heat-inactivated serum over 48 h at 37 °C. At defined time intervals, i.e., 1, 3, 6, 24, and 48 h, aliquots were collected and stored at −20 °C till gel electrophoresis was executed. 

### 2.4. Cells 

The immortalized rat striatal neuronal cell line ST14A was purchased from Coriell Institute (Camden, NJ, USA) and cultured in DMEM with 4.5 g/L glucose, L-glutamine, and sodium pyruvate with 10% fetal bovine serum (FBS) and 1% penicillin/streptomycin. Cells were maintained at 33 °C, with a 5% CO_2_ atmosphere, and the cell culture medium was changed every 2–3 days. The ST14A cell line expresses human mHTT (residues 1-548) with 120 polyglutamines/CAG repeats (Q120).

The immortalized human cerebral microvascular endothelial cell line (hCMEC/D3) was purchased from Cedarlane Laboratories (Burlington, Ontario, Canada) and cultured on rat collagen I-coated flasks (CELLCOAT^®^, Greiner Bio-One, Kremsmünster, Austria) and maintained in Endothelial Growth Medium (EBM-2, #CC-3156, Lonza, Basel, Switzerland) supplemented with SingleQuots™ Bullet-Kit (#CC-4176, Lonza, Basel, Switzerland) at 37 °C in 5% CO_2_. The growth medium was changed every 2–3 days until the cells reached approximately 80% confluency. The cells used were between passages 27 and 35.

Human HD patient-derived fibroblasts GM04723 (19-year-old, female, CAG15/67) were purchased from Coriell Institute and maintained in complete minimal essential medium with 15% FBS, 1% L-glutamine, and 1% penicillin/streptomycin at 37 °C in 5% CO_2._ The cell culture medium was changed every 2–3 days. The cells were cultured for approximately 20 passages. 

#### In Vitro Blood–Brain Barrier (BBB) and Co-Culture Model

The hCMEC/D3 BBB cell model was previously established and characterized in our laboratory [16]. Briefly, hCMEC/D3 cells were seeded on coated transwell inserts (ThinCert #662641, Greiner Bio-One, Kremsmünster, Austria) at a density of 3.0 × 10^4^ cells/cm^2^ in 150 µL complete medium and grown for 8 days to achieve confluence (TEER plateau of 20 Ω × cm^2^). After the establishment of the hCMEC/D3 monolayer, the upper chamber was transferred to another bottom chamber seeded with ST14A cells (7.5 × 10^4^ cells per well in 600 µL of DMEM media) or GM04723 (1.5 × 10^5^ cells per well in 600 µL of DMEM media) the day before. Then, free ASO or CD3:ASO conjugated to DSPE-PEG or co-formulated with RVG in EBM-2 medium were added to the upper compartment. After incubation for 72 h, ST14A and GM04723 cells in the lower chamber were collected for western blot analysis and RT-qPCR, respectively.

### 2.5. Viability Assay

The cell viability was assessed by ATP content using the CellTiter-Fluor™ cell viability assay (Promega, Madison, WI, USA). ST14A cells (7.5 × 10^4^ cells per well) and GM04723 (1.5 × 10^5^ cells) were seeded in 96-well plates and cultured for 24 h, whereas hCMEC/D3 cells (3.0 × 10^4^ cells per well) were seeded in coated 96-well plates and cultured for 8 days. After this period, cells were treated with CDs:ASO nanocomplexes for 72 h, and the cell viability from each treated sample was measured in triplicate according to the manufacturer’s instructions.

### 2.6. Cellular Uptake

The amount of fluorescently labeled 6-FAM ASO taken up by cells was measured by flow cytometry and confocal microscopy. ST14A cells were plated in 24-well tissue culture plates at 1.5 × 10^5^ cells/well 24 h prior to transfection. After 24 h, cells were washed twice with D-PBS and CDs:ASO nanocomplexes were added to cells in serum-reduced medium (2% FBS) at a final concentration of 100 nM 6-FAM labeled ASO. After 6 h of incubation, cells were washed with D-PBS, trypsinized, and collected into a complete DMEM medium. Samples were centrifuged for 5 min at 1500 rpm at 4 °C and resuspended in 400 μL of D-PBS. All data were acquired on a BD FACSCelesta™ flow cytometer (FACSCalibur, BD Biosciences, San Jose, CA, USA) and analyzed using FlowJo™ Software (version 10).

For confocal imaging, ST14A cells (1.25 × 10^5^ cells per well) and GM04723 (1.0 × 10^5^ cells) were seeded on round coverslips, at per coverslip in a 24-well plate and 500 uL medium. After 24 h, the medium was replaced with 400 uL of serum-reduced medium (2% FBS) plus 100 uL of 6-FAM labeled ASO (at a final concentration of 100 nM ASO). For staining of lysosomes, 50 nM LysoTracker Red DND-99 (ThermoFisher, Bremen, Germany) was added into the medium 1 h prior to fixation with 4% PFA (6 h of incubation in total). Coverslips were mounted in Mowiol, and confocal images were acquired on a Zeiss LSM510 META confocal microscope (Carl Zeiss, Jena, Germany) using a 63× objective.

### 2.7. Protein Quantification

ST14A cells were seeded in 24-well plates at a density of 7.5 × 10^4^ cells per well and incubated for 24 h. After growing overnight, the medium was replaced with 400 uL serum-reduced medium (2% FBS) plus 100 µL of CDs containing ASO (at a final concentration of 100 nM ASO). For co-culture experiments, ST14A cells were treated as described above.

#### 2.7.1. Western Blotting

Cells were lysed 72 h post-transfection in RIPA buffer (Sigma Aldrich, Wicklow, Ireland) containing 200 mM PMSF, 100 mM sodium orthovanadate, and cOmplete™ EDTA-free protease inhibitor cocktail solution (Roche, San Francisco, CA, USA) for 30 min on ice. The lysate was centrifuged at 14,000 rpm at 4 °C for 10 min and protein concentration was quantified using Pierce™ BCA Protein Assay kit (ThermoFisher, Bremen, Germany). The samples were mixed with an equal volume of Laemli buffer (Sigma Aldrich) and denatured by boiling at 95 °C for 6 min. Equal amounts of protein (15 µg) were loaded into a 6–12% SDS-PAGE gel and then transferred to PVDF membranes (Bio-Rad, Hercules, CA, USA). Membranes were blocked with 5% milk in TBST buffer (150 mM NaCl; Tris, pH 7.4 and 0.1% Tween-20) for 1 h at room temperature, rinsed, and incubated overnight at 4 °C with primary antibody against HTT (1:1000; MAB2166; Sigma Aldrich, Wicklow, Ireland). Subsequently, the membranes were incubated with anti-mouse HRP-conjugated secondary antibodies for 2 h at room temperature (1:2000; A0545; Sigma Aldrich, Wicklow, Ireland). The chemiluminescent signal was revealed using SuperSignal™ West Pico PLUS Substrate (ThermoFisher, Bremen, Germany) and imaged with an Alliance Imaging system (Uvitec, Cambridge, UK). Band intensities were quantified by densitometry using the LabImage *1D* program and normalized to β-actin control (1:1000; A5441; Sigma Aldrich, Wicklow, Ireland).

#### 2.7.2. Immunohistochemistry

Cells were seeded in coverslips and transfected for 72 h. Then, the cells were washed with warm D-PBS, fixed with 4% paraformaldehyde (PFA) at room temperature for 15 min, and quenched with 50 mM ammonium chloride followed by permeabilization with 0.2% Triton X-100 for 5 min. Blocking was performed with 3% BSA for 30 min at room temperature. Afterward, the samples were incubated with anti-HTT antibody, clone 1 HU-4C8 (1:100; MAB2166; Sigma Aldrich, Wicklow, Ireland), and Anti-polyQ specific, clone MW1 (1:100; MABN2427; Sigma Aldric, Wicklow, Ireland) overnight at 4 °C. Following incubation, the samples were washed in D-PBS and incubated for 1 h in 1% BSA with Alexa 488 goat anti-mouse (1:200; #A11001; Invitrogen, Waltham, MA, USA) and DAPI (0.1 µg/mL, # D1306; Invitrogen, Waltham, MA, USA). Finally, samples were washed in D-PBS, mounted with ProLong™ Gold anti-fade reagent (ThermoScientific, Bremen, Germany), dried, and imaged using an Olympus IX73 microscope (Olympus Inc., Tokyo, Japan). Immunohistochemistry was carried out at the same time to allow for a comparison of the mean fluorescence intensity (MFI), which was quantified using Image J software (NIH) (version 1.53t).

### 2.8. SNP Genotyping Assay

GM04723 (CAG15/67) cells were seeded in 24-well plates at a density of 1.5 × 10^5^ cells per well and incubated for 24 h. After growing overnight, the medium was replaced with 400 uL serum-reduced medium (2% FBS) plus 100 µL of CDs containing ASO (at a final concentration of 100 nM ASO). RNA extractions were performed 72 h post-transfection using a GenElute™ Mammalian Total RNA kit according to the supplier’s instructions (Sigma Aldrich, Wicklow, Ireland). cDNA synthesis was further performed using the High-Capacity cDNA Reverse Transcription kit (Applied Biosystems, Foster City, CA, USA). Allele-specific detection of *HTT* mRNA levels was performed using the SNP rs362307 (C___2229302_10, Thermo Fisher Scientific, Bremen, Germany) genotyping assay. For each sample, 10-μL qPCR reactions were set up as follows: 5 µL SsoFast Probes Supermix (Bio-Rad, Basel, Switzerland), 0.25 µL of 40× TaqMan SNP Genotyping Assay, 2.75 µL H_2_O, and 2 µL of cDNA sample. All reactions were carried out using the CFX96 Real-Time PCR Detection System (Bio-Rad, Basel, Switzerland) following the steps: 95 °C for 30 s; 95 °C for 5 s; 60 °C for 25 s; 40 cycles of amplification. Each reaction was run in duplicate. Results were expressed as Ct values after normalization to GAPDH as the housekeeping gene. 

### 2.9. Statistical Analysis

All statistical analyses were performed using GraphPad Prism 7. Analysis of protein knockdown was performed using two-way ANOVAs with Sidak’s post hoc multiple comparison testing. Immunohistochemistry and RT-qPCR experiments were analysed using one-way ANOVA with Tukey’s post hoc multiple comparison testing. All error bars represent the standard error of the mean. Statistical significance was considered when *p* < 0.05.

## 3. Results

### 3.1. Physicochemical Characterization of CDs:ASO Nanocomplexes

Initial studies were performed to determine the optimal mass ratio, CDs:ASO, regarding the size, PDI, zeta potential, and binding. The optimal mass ratio between CDs and ASO was influenced by the CDs chemical structure. A mass ratio higher than 5:1 was suitable for CD1:ASO and CD3:ASO nanocomplexes formation. However, when the CD2:ASO nanocomplexes were prepared at a mass ratio lower than 7.5:1, no efficient NPs formation could be observed and a free ASO signal was detected by agarose gel electrophoresis (Appendix A). For this reason, the mass ratio of 10:1 was chosen for all CDs.

The physicochemical characterization of the CDs:ASO nanocomplexes is shown in Figure 1. Morphological characterization by TEM illustrated that the nanocomplexes were spherical (Figure 1A), with sizes of 106 nm for CD1:ASO, 127 nm for CD2:ASO, and 124 nm for CD3:ASO. DLS intensity histograms show that the nanocomplexes have narrow size distributions, with sizes ranging from 148–175 nm (Figure 1B). 

### 3.2. Biocompatibility Evaluation

Biocompatibility is a primary requirement in the development of delivery systems for clinical use. Therefore, the potentially toxic effects of CDs:ASO nanocomplexes on ST14A cells were evaluated by measuring ATP levels. The three nanocomplexes were non-toxic at the concentration and time evaluated, with cell viability exceeding 92% (Figure 2A). To mimic the in vivo environment, the CDs:ASO nanocomplexes were incubated in a solution containing 25% (*v*/*v*) heat-inactivated serum at 37 °C for different periods (Figure 2B). The results indicated that both free ASO and CDs:ASO nacomplexes were protected against serum nucleases for up to 48 h (Figure 2B), probably due to the heavy chemical modification of the ASO. Although not degraded by the nucleases, the net negative charge of the free ASO will impede permeability across the negatively charged cell membrane, thus inhibiting cell entry. To address this hypothesis, cellular uptake of the CDs:ASO nanocomplexes in ST14A cells was evaluated.

### 3.3. Cellular Uptake

FAM-labeled ASO alone or FAM-labeled ASO loaded into CDs was incubated with the cells for 6 h and the levels of uptake were determined by flow cytometry (Figure 3A) and confocal microscopy (Figure 3B). Free ASO displays weak cellular uptake (1.06%), which thus limits therapeutic applications. In contrast, CDs:ASO nanocomplexes were efficiently taken up by cells. The chemical structure of the delivery system significantly affected the ASO uptake, in the order CD3:ASO (88.3%) > CD1:ASO (65.7%) > CD2:ASO (21.5%) (Figure 3A). 

In the absence of CDs, no cell-associated fluorescence was observed (Figure 3B), corroborating that ASO itself cannot be efficiently internalized by ST14A cells. All FAM-labeled CDs:ASO nanocomplexes showed cellular uptake; however, the fluorescence subcellular intensities and distribution patterns were very different. Both CD1:ASO and CD3:ASO exhibited perinuclear enrichment, with CD3:ASO showing a visibly increased green fluorescence signal, which is consistent with the flow cytometry results. CD2:ASO labeled just a few punctate structures localized at the periphery of the cell.

### 3.4. Efficacy in a Neuronal In Vitro Model of HD

The silencing activity of CDs:ASO nanocomplexes was tested at the protein level in ST14A cells, which express an N-terminal 548-amino acid fragment of mHTT containing 120 polyglutamine repeats and has been established as a reliable model of HD [18,19]. As shown in Figure 4, both CD1:ASO and CD3:ASO nanocomplexes were able to efficiently downregulate the mHTT and the endogenous wtHTT protein; however, the reduction of the mHTT levels was greater than that of wtHTT. Comparing the three CDs, the treatment with CD3:ASO nanocomplexes showed a more robust activity, reducing the levels of mHTT to 46% (*p* < 0.001) versus the reduction to 68% (*p* < 0.05) induced by CD1:ASO. A similar pattern was observed for wtHTT; CD3:ASO reduced the levels of wtHTT to 58% (*p* < 0.01) while CD1:ASO downregulated the levels to 74%. No significant difference between mHTT and wtHTT expression was detected in the cells treated with CD2:ASO. This difference in activity is probably linked to the later release of free ASO by CD2 (Appendix A).

To confirm the efficacy and specificity of CDs:ASO nanocomplexes, immunohistochemistry using a polyQ (mutant) and a non-polyQ targeting antibody were performed. The non-polyQ antibody MAB2166 (clone 1HU-4C8) was selected because it can detect total HTT (wild type and mutant forms), whereas the polyQ antibody MABN2427 (clone MW1) was chosen since it is the most widely used polyQ directed antibody. Furthermore, MW1 is currently being employed for mHTT detection in clinical studies aimed at lowering mHTT levels [23].

ST14A striatal cells showed similar cytoplasmic labeling for both antibodies (Figure 5), although the mean fluorescence intensity of the mutant antibody was stronger than that of total HTT (9.23 ± 1.05 vs. 6.33 ± 0.77, respectively). Moreover, the extent of mHTT/total HTT protein silencing was greater than evidenced by western blotting. Compared with the control group, CD3:ASO treatment significantly reduced 90% of mHTT levels (*p* < 0.001), while reducing around 55% of total HTT (*p* < 0.05). Similar trends were observed for CD1:ASO, as mHTT expression was inhibited by approximately 67% (*p* < 0.01) and total HTT by 59% (*p* < 0.01). CD2:ASO nanocomplexes showed no significant effects on mHTT and total HTT expression.

### 3.5. Improving the Delivery and Efficacy of the Best-Performing Nanocomplex (CD3:ASO) with RVG Targeting 

One of the biggest challenges in the nucleic acid therapeutic field has been targeted delivery to the central nervous system (CNS). To provide a strategy that improves the CNS delivery of CD3:ASO nanocomplexes, an RVG-targeted PEGylated CD-based formulation was developed. The surface modification of CDs:ASO was achieved using the postinsertion technique. The DLS results showed that CD3:ASO:DSPE-PEG (without RVG addition), herein referred to as -RVG or untargeted, and CD3:ASO:DSPE-PEG-RVG, herein referred to as + RVG or RVG-targeted, demonstrated larger particle sizes (169 ± 6.06 nm and 172 ± 5.49 nm, respectively) compared to CD3:ASO (148 ± 3.33 nm), indicating the successful incorporation of DSPE-PEG and DSPE-PEG-RVG. The PDI of untargeted and RVG-targeted nanocomplexes was 0.20 ± 0.013 and 0.16 ± 0.013, respectively, indicating the generation of monodisperse formulations. Additionally, the surface charge of untargeted nanocomplexes remained similar to CD3:ASO (~16 mV), while the surface charge of the RVG-targeted counterparts decreased to 9 ± 0.81 mV.

The morphological analysis by TEM showed that all the formulations consisted of spherical-shaped nanoparticles, with no apparent differences between RVG-targeted and non-targeted CD3:ASO nanocomplexes (Appendix A). The preparation of the nanocomplexes in a cell culture medium, essential for further co-culture experiments, did not significantly modify the average size of both formulations, with diameters around 170 nm and PDIs lower than 0.25 (Appendix A). The size stability of untargeted and RVG-targeted nanocomplexes was also monitored for 3 days at 37 °C. Only marginal changes (±10 nm) were observed (Appendix A). Overall, both formulations remained well dispersed without substantial aggregation in the protein-rich medium when stored at 37 °C. Furthermore, the addition of RVG displayed negligible cytotoxicity, with cell viability above 86% relative to untreated ST14A cells (Appendix A).

The efficiency of RVG-targeted nanocomplexes in delivering ASO was evaluated by monitoring their cellular uptake and endo-lysosomal escape, since ASO needs to be released into the cytoplasm for efficient gene silencing. Confocal microscopy allowed a clear view of the internalization of the RVG-targeted nanocomplexes inside the ST14A cells. As shown in Figure 6, the green fluorescence signal was predominantly visualized in the cytoplasm, with a slightly stronger fluorescence detected for the RVG-targeted than the untargeted nanocomplexes. No green signal was detected in the cells treated with ASO alone. Late endosomes and lysosomes were labeled with LysoTracker Red. As the images illustrated, only a very small amount of the ASO overlapped with the red fluorescence of late-endo/lysosomes 6 h post-transfection, indicating that both nanocomplexes escape from endo-lysosomes to the cytoplasm after internalization.

Next, the effect of surface modification by active targeting on the ability of CD3:ASO nanocomplexes to cross the in vitro BBB model and induce silencing in the targeted neuronal cell was investigated, as schematically shown in Figure 7A. After 72 h of transfection, untargeted nanocomplexes failed to induce silencing of both wtHTT and mHTT, while RVG-targeted nanocomplexes significantly downregulated the expression of mHTT compared to control (*p* < 0.001) (Figure 7A), indicating that RVG-targeted nanocomplexes efficiently permeate through the monolayers and diffuse to reach the underlying cell layer. Given that the expression of the wtHTT was not reduced, this result confirmed the allelic preference of RVG-targeted nanocomplexes in this model. No apparent cytotoxicity was observed in hCMEC/D3 cells, with cell viability exceeding 85% (Appendix A) and similar TEER values between the treated and non-treated cells (Appendix A).

The wtHTT and mHTT levels after direct incubation of ST14A cells alone (monoculture) with RVG-targeted and untargeted nanocomplexes were also evaluated (Figure 7B). The reduction of wtHTT and mHTT expression was higher in the monoculture ST14A cells than in the co-culture model, indicating that the BBB represents a formidable barrier even for small molecules such as CDs:ASO. As shown in Figure 7B, both RVG-targeted and untargeted nanocomplexes were able to downregulate the wtHTT and mHTT protein levels in a similar manner. The treatments reduced mHTT protein by ~50% (*p* < 0.001) and wtHTT protein by ~30% (*p* < 0.01) compared to control. 

Additionally, the interaction between the variables (targeting and mutation) was evaluated (Appendix A). The analysis revealed that the monoculture and co-cultured cells were significantly affected by targeting and mutation. However, the interaction between targeting and mutation was significant only in the co-culture model (*p* < 0.01).

### 3.6. HD Patient-Derived Fibroblasts 

To further corroborate our data, we evaluated the effectiveness of the CD3:ASO nanocomplexes in reducing allele mRNA in the HD patient-derived fibroblast cell line GM04723 (CAG15/67), which encompasses a wide variety of HD patient genotypes [24]. To distinguish the normal and disease HTT alleles in HD patient-derived cells, the heterozygous SNP rs362307 in exon 67 (SNP1), notably observed in approximately two-thirds of HD patients, was evaluated [25]. Firstly, the top-performing CD was confirmed through a screening phase. Flow cytometry analyses revealed that free ASO alone exhibited low cellular uptake (1.51%) into HD patient-derived fibroblasts after 6h of incubation. Conversely, the CDs displayed relatively higher cellular uptake levels: 75.3%, 33.7%, and 88.3% of ASOs for CD1:ASO, CD2:ASO, and CD3:ASO nanocomplexes, respectively (Figure 8A).

We next investigated whether the differences in ASO cellular uptake directly correlated with mRNA silencing efficacy. As shown in Figure 8B, cells transfected with CD1:ASO and CD3:ASO showed a decrease in SNP1 mRNA levels by 34 and 41% (*p* < 0.05), respectively. The lower mRNA silencing efficacy of CD2:ASO nanocomplexes (27% reduction) was likely attributed to the lower ASO uptake, demonstrating a correlation between ASO cellular uptake and activity. Importantly, cellular viability was not affected by ASO treatment, with cell viability exceeding 92% (Figure 8C).

The lead formulation—CD3:ASO—was then conjugated to RVG and the efficacy of RVG-targeted nanocomplexes in delivering ASO was evaluated by monitoring the cellular uptake and endo-lysosomal escape. After 6 h incubation, fibroblasts showed efficient uptake of both RVG-targeted and untargeted nanocomplexes, as indicated by the strong ASO green signal distributed in the cytosol (Figure 9). More importantly, the green signal barely overlapped with the red ones (late endo-lysosomes), demonstrating that the majority of the ASO escaped from the late-endo/lysosomes to the cytosol, which, thereby, would lead to efficient gene silencing. 

Gene silencing was evaluated in HD fibroblasts cultured in monoculture and co-culture with hCMEC/D3 cells (BBB model), (Figure 10). In line with the experiments with the striatal neuronal cells, RVG-targeted nanocomplexes performed best in the co-culture model, reaching about 37% (*p* < 0.001) downregulation of SNP1 levels. Non-targeted nanocomplexes showed similar results to free ASO, with about a 13% (*p* < 0.05) reduction in SNP1 levels. These experiments confirmed the efficient delivery of ASO by CD3 in a challenging cell culture system, which more closely mimics the in vivo conditions.

In the monoculture system, RVG-targeted and non-targeted groups showed a similar trend (Figure 10B). Both showed a >45% (*p* < 0.001) reduction in SNP1 mRNA levels compared to the control, confirming that the presence of RVG in the formulation did not result in higher efficacy in fibroblasts. No apparent cytotoxicity associated with CDs:ASO nanocomplexes was observed (Appendix A).

## 4. Discussion

Therapeutic strategies focused on lowering mHTT levels using non-allele and allele-selective ASOs delivered by intrathecal injections have been assessed in human clinical trials; however, the three most promising HTT-lowering trials were halted early [8]. Although they have failed for very different reasons (reviewed elsewhere [26]), a common potential challenge for this therapy is the long-term tolerability of the monthly intrathecal injections [5]. Therefore, the exploration of alternative strategies that can deliver ASOs to the brain in an effective way and using lower doses has become an important aspect of the development of RNA-based therapeutics.

In this study, a different approach was used to lower total HTT expression using ASO delivered via CDs. The three CD-derivatives have been all modified on the primary side with linear dodecanthiol moieties in order to provide a common lipophilic motif utilizable for the self-assembly process. CD1 possesses alkyl chains on the secondary side terminating with primary amines. A suitable precursor of CD1 has been regioselectively modified on the hydroxy in position 2 with propargyl groups that have been reacted with BOC-protected 1-azidopropylamine under cycloaddition conditions. Acidic cleavage of the BOC protecting groups yielded the desired compound. The installation of the tertiary amines on the secondary side of CD2 was achieved through ester formation between the commercially available 3-aminobutanoyl chloride and the corresponding primary-side modified cyclodextrin precursor. The resulting isomeric mixture is primarily substituted on the hydroxy groups in position 2. Exhaustive secondary side propargylation for CD3 can be achieved as previously described [27]. Subsequent cycloaddition with ad hoc designed azido-alkyl-amine derivatives generated the titled compound in good yield. CD3 possesses alkyl chains on the secondary side, terminating with primary amines as with CD1, but in a larger amount (more than double). These structure modifications of CDs can directly affect their interactions with cellular components and, consequently, their biological activity [28,29,30]. The physicochemical properties of the three CDs evaluated were similar and with uniform distribution, showing desirable characteristics for brain delivery (<200 nm and positively charged) [31], strong stability, and protecting ASO from nuclease degradation. In the cytotoxicity evaluation, it was proven that CD1:ASO, CD2:ASO, and CD3:ASO nanocomplexes were not toxic to striatal neuronal cells nor HD patient-derived fibroblasts under the conditions used. 

Given the complexity of HD, it is important to validate the efficacy of the delivery system in different therapeutically relevant cell lines. Striatal neurons are particularly vulnerable at the early stages of HD progression [32], while in human fibroblast lines, heterozygosity of HD-SNPs provides a convenient and physiologically relevant platform for testing potential allele-selective ASOs. In the current study, we showed greater cellular uptake and silencing efficacy by treatment with double-charged chain γ-CDs (CD3:ASO nanocomplexes) than β-CDs with primary amines (CD1) followed by β-CDs with tertiary amines (CD2). Although the ASO sequence was not designed to target specific polyQ repeats within mHTT mRNA, a higher inhibition of the expression of mHTT protein was achieved compared to total HTT. 

We hypothesized that active targeting would increase nanocarrier internalization in the targeted cells, thereby increasing the ASO intracellular concentration and efficacy. For this reason, CDs were further modified with RVG, a cell-penetrating peptide that has been considered an attractive approach for the targeted transport of nanocarriers across the BBB [17,33]. After incorporating RVG into γ-CDs, clear differences were observed between the mono- and co-culture models. The presence of the targeting moiety did not result in preferential accumulation in the monoculture system compared to the co-culture model, probably because it cannot facilitate the critical step of extravasation through the BBB. Although nAChR is expressed in endothelial cells, neurons, and fibroblasts, nAChRs in the CNS comprise a combination of different α (α1–α9) and β (β2–β4) subunits, or even α subunits alone. Depending on the cell type involved, the diversity in expression, dominance, and subunit composition may have different effects on ligand affinity [34,35]. Considering that nAChR was responsible for RVG uptake, in order to cross the BBB, it would have to bind to either α3, α5, α7, β2, or β3 subunits, whereas striatal neurons would have to bind to either α3, α4, α7, or β2 [36]. It is reasonable to assume that the synthesized RVG sequence selected was responsible for interaction mainly with the BBB nAChR subunits and the relatively similar size and positive zeta potential of both RVG-targeted and untargeted nanocomplexes were accountable for their comparable efficiency in the monoculture model. 

Although this study has provided a foundation for the rational design and future application of CD platforms in the delivery of ASOs to treat HD, some limitations still need to be addressed in order to successfully translate these findings into the clinic. For example, preclinical studies of allele-specific targeting ASOs should be performed in animal models with the same SNP genotypes as the prospective patient population. However, there are no mouse models of HD that include SNP1 and SNP2 [37]. This was one of the challenges encountered by Wave Life Sciences in their WVE0120101 and WVE-120102 preclinical trials, as they needed access to suitable animal models of HD that contained the SNPs it was targeting. Therefore, they were unable to confidently predict the pharmacokinetic and pharmacodynamic effects of the molecules [38].

## 5. Conclusions

In this work, we have explored the influence of distinct chemical structures on the biological behavior of CDs-based nanoparticles for the delivery of ASO. The double-charged chain γ-CDs exhibited superior internalization and knockdown efficacy compared to both β-CDs. Additionally, all three CDs evaluated were found to be non-cytotoxic in striatal neurons and HD fibroblast cells. Although designed as a non-selective approach, our strategy provides an additional benefit of some allele selectivity as demonstrated by western blotting and immunohistochemistry against the mutant polyQ tract. After the incorporation of RVG into γ-CDs, clear differences were observed between the mono- and co-culture models. RVG-targeting did not result in preferential accumulation of the nanocomplexes in the neuronal/fibroblast monoculture but did so in the co-cultured BBB model. These results may be an indication that, by direct injection into the brain, the use of untargeted CDs:ASO nanocomplexes might be sufficient to induce gene/protein knockdown. However, if systemic administration is desired, a targeting approach should be considered to overcome the BBB. Further studies will now be performed in vivo by intrathecal delivery as well as by intravenous delivery to corroborate these data.

## Data Availability

The majority of the data are presented in this publication. Additional data will be provided by the authors upon reasonable request.

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
