# Peer review of "Cyclodextrin-Based Nanoparticles for Delivery of Antisense Oligonucleotides Targeting Huntingtin"

_pharmaceutics, 2023, doi:10.3390/pharmaceutics15020520_

Round 1
Reviewer 1 Report
Authors present a cyclodextin approach for cell-delivery int he context of Hungtington disease. The study is solid and analyses reasonably accurate.
Few comments:
-check for typos
-section 2.7 Protein quantification 233 ST14A cells were s doesnt mention the protein quantification assay used
-supplementary figures have axes reaching 150%, is this feasible? see also fig 8 and 10 for example.
Reviewer 2 Report
The manuscript "Cyclodextrin-based nanoparticles for delivery of antisense oligonucleotides targeting Huntingtin" Mendonça M. et al. is devoted to an actual problem - targeted delivery of antisense RNA to cells expressing mutant huntingtin containing the glutamine tract. Three variants of nanoparticles based on cyclodextrin, which form complexes with antisense RNA to glutamine tract were fabricated. The complexes were resistant to extracellular RNases and do not exhibit cytotoxicity. The complexes are taken up by cells and inhibit the synthesis of mutant huntingtin, the effect being reproduced in cultured fibroblasts derived from patient with HD. The authors use an adequate set of modern methods. Familiarization with the work leaves a very pleasant impression, but one wonders, why HD mouse models are not used in the work. However, at the end of the article, the authors explain their plans. So, there are no comments on the work, and it can be published in its present form.
Author Response
We thank you for the time and effort you spent reviewing our manuscript. We are pleased with the positive evaluation of our study.
Reviewer 3 Report
Mendonca and colleagues developed an improved method of delivery of antisense oligonucleotides (ASO) targeting Huntingtin to neurons. ASO was delivered in cyclodextrin (CD) nanoparticles to a co-culture of endothelial cells (in upper chamber) and neurons (in lower chamber). The delivery of ASO resulted in a significant decrease of Huntigtin expression. gamma-CD were more efficient than beta-CD. The delivery of ASO and suppression of Huntingtin levels were increased by incorporation of the brain-targeting peptide RVD in CD nanoparticles. This is a solid study opening novel perspectives for the development of the methods of Huntingtin disease treatment.
Author Response

(The authors gave the same response as above.)
